# A spider mating plug functions to protect sperm

He Jiang[1], Yongjia Zhan[1], Qingqing Wu[2], Huitao Zhang[3], Matjaž Kuntner[4,5,6,7], Lihong Tu[1]*

1 College of Life Sciences, Capital Normal University, Beijing, P. R. China, 2 Lang Yue Campus of Beijing 12th High School, Beijing, P. R. China, 3 Beijing Advanced Innovation Center for Imaging Technology, Capital Normal University, Beijing, P. R. China, 4 Department of Organisms and Ecosystems Research, National Institute of Biology, Ljubljana, Slovenia, 5 Jovan Hadži Institute of Biology, ZRC SAZU, Ljubljana, Slovenia, 6 State Key Laboratory of Biocatalysis and Enzyme Engineering & Centre for Behavioural Ecology & Evolution, School of Life Sciences, Hubei University, Wuhan, China, 7 Department of Entomology, National Museum of Natural History, Smithsonian Institution, Washington, DC, United States of America

* tulh@cnu.edu.cn

**Data Availability Statement:** All relevant data are within the manuscript and its Supporting Information files.

**Funding:** Funding for this research is provided by the National Natural Sciences Foundation of China NSFC-31572244 and NSFC-31872188 to LT,

## Abstract

Mating plugs in animals are ubiquitous and are commonly interpreted to be products of mating strategies. In spiders, however, mating plugs may also take on functions beyond female remating prevention. Due to the vagaries of female genital (spermathecal) anatomy, most spiders face the problem of having to secure additional, non-anatomical, protection for transferred sperm. Here, we test the hypothesis that mating plugs, rather than (or in addition to) being adaptations for mating strategies, may serve as sperm protection mechanism. Based on a comparative study on 411 epigyna sampled from 36 families, 187 genera, 330 species of entelegyne spiders, our results confirm the necessity of a sperm protection mechanism. We divided the entelegyne spermathecae into four types: SEG, SED, SCG and SCD. We also studied detailed morphology of epigynal tracts in the spider *Diphya wulingensis* having the SEG type spermathecae, using 3D-reconstruction based on semi thin histological series section. In this species, we hypothesize that two distinct types of mating plug, the sperm plug and the secretion plug, serve different functions. Morphological details support this: sperm plugs are formed on a modified spermathecal wall by the spilled sperm, and function as a temporary protection mechanism to prevent sperm from leaking and desiccating, while secretion plugs function in postcopulation both as a permanent protection mechanism, and to prevent additional mating. Furthermore, with the modified spermathecal wall of $S_2$ stalk, the problem of shunt of sperm input and output, and the possibility of female multiple mating have been resolved. Variation in spermathecal morphology also suggests that the problem of sperm protection might be resolved in different ways in spiders. Considering mating plugs of varying shapes and origins in the vast morphospace of spiders, we conclude that mating plugs might serve different purposes that relate both to mating strategies, as well as to sperm protection.

NSFC-61671311 to HZ and by Slovenian Research and Innovation Agency P1-0255 and J1-50015 to MK. The funders had no role in study design, data collection and analysis, decision to publish, or preparation of the manuscript.

**Competing interests:** the authors have declared that no competing interests exist.

**Abbreviations:** ACD, paired CD included in an atrium; ACG, paired CG included in an atrium; ASCP, single SCP blocking atrium completely; ASCP-inc, single SCP blocking atrium incompletely; AT, atrium; CD, copulatory duct; CG, copulatory groove; CGD, CG+CD; CGDG, CG+CD +CG; CGF, CG slit on EP fused; CGSCP, SCP in CG slit; CGSP, SP in CG slit; CO, copulatory opening; COGP, GP in CO; COSCP, SCP in CO; COSP, SP in CO; CPG, pocket-like CG; CT, copulatory tract; EP, epigynal plate; EPSCP, SCP on EP coving over both SG slits and CO; EPSP, SP on EP; ET, epigynal tract; FD, fertilization duct; FD+CFD, paired FD convergent into common FD; FDG, FD+FG; FDG +CFG, paired FD+FG convergent into common FG; FG, fertilization groove; FG+AFD, FG with additional FD; FG-OEF, FG slit ending outside epigastric furrow; FT, fertilization tract; GC, groove cavity; GE, groove edge; GP, genital plug; HCO, CO hidden; S, spermatheca; S1, primary spermatheca; S2, secondary spermatheca; Scape, EP protruding and prolonged, carrying CG out, CO located at distal part of scape; SCD, spermatheca of compacted duct; SCG, spermatheca of compacted groove; SCP, secretion plug; SD, spermathecal duct; SED, spermatheca of elongated duct; SEG, spermatheca of elongated groove; SG, spermathecal groove; SGF, SG slit on EP fused; SGSCP, SCP in SG slit; SGSP, SP in SG slit; SP, sperm plug; SP+SCP, SP and SCP coexist; SSCP, SCP at spermathecal entrance; ST, spermathecal tract.

## Introduction

An interesting phenomenon commonly found in many animal groups is that mating plugs of various forms left in female copulatory tracts after copulation, usually function as copulatory barriers to lower female remating chances. As such, mating plugs are generally interpreted to represent a male mating strategy [1–5]. In spiders, however, particularly in entelegyne lineages, mating plugs may also serve other functions beyond a male mating strategy [6, 7]. An internal fertilization mechanism in spiders dictates the copulation process be separated from fertilization. This allows for female polyandry prior to oviposition, which in turn intensifies sexual selection and facilitates a diversity of shapes and forms in mating strategies, not only for males, but also for females [6, 8]. Female spermathecae store sperm for prolonged times between copulation and fertilization [8–10] and this necessitates some sort of a sperm protection mechanism [11]. Mating plugs are ubiquitous in spiders, and are usually explained to function as a product of male, or female mating strategies (the 'mating strategy hypothesis') [6, 8, 12–14] since they provide physical blockage that prevents additional mating. However, mating plugs could plausibly also provide sperm protection (the 'sperm protection hypothesis') [7, 11]. These two functional hypotheses, of course, need not be mutually exclusive.

By far most species of spiders are described as entelegyne, meaning that their copulatory and fertilization ducts to and from the spermathecae are unlinked [15]. In stark contrast to the alternative condition (haplogyne), the entelegyne gestalt poses a unique challenge related to sperm protection. Namely, the entelegyne spermatheca has separate openings for sperm inflow and outflow, respectively, and strongly sclerotized walls of these openings render the spermathecal entrances perpetually open [7, 15, 16]. Furthermore, spermathecae store sperm for extended periods of time only to be released during oviposition [8–10]. These facts pose certain risks for the viability of sperm lodged in spermathecae: sperm leakage and desiccation during post copulation, and sperm backflow when being released. Structural features to avoid these risks, such as potential valves that would close the spermathecal entrances, would be logically expected, but we know little about it.

A recent study on the spider *Holocnemus pluchei* found [17] that the viability of sperm stored in female genital tracts decreased compared to that in male and that this decrease was not directly related to the sperm storage time. Interestingly, stored sperm from the males having performed a longer post-insemination behavior retained a higher sperm viability. While it seems that female storage negatively affects sperm viability [17, 18], the reasons are not entirely clear. There are multiple possibilities, such as the quality of the sperm itself [19], the issue of sperm aging [20], and the trade-off between the female immune system and maintaining stored sperm motility [21], as well as sperm competition or cryptic female choice that take place in the female genital tracts [17, 22–25]. Nevertheless, it cannot be ignored that the inherent defects in the structure of the entelegyne spermathecae pose the risk of leakage and desiccation to the stored sperm. The decrease in sperm viability within the female tracts did not worsen over time, indicating effective risk control. Among the possible male post-insemination behavior, e.g. acting as post-copulatory mate guarding [26–28] or copulatory courtship [29–32], dedicating to generate mating plugs [7, 33] might be the most likely help to maintain sperm viability. Huber [11] proposed that mating plugs may be a potential sperm protection mechanism, but there has been a lack of empirical evidence (but see [7]).

Previous studies of mating plugs in spiders have not reached a general conclusion as to their function. Mating plugs refer to either genital pieces (referred as genital plug (GP)) or hardened amorphous materials, mainly formed by sperm or secretions (referred as sperm plug (SP) or secretion plug (SCP), [7]). Although mating plugs were also recorded in some haplogyne groups, e.g. pholcid spiders [34, 35], they are more commonly found in entelegyne spiders.

Lodged in copulatory openings during or after copulation, they are generally thought to function as a mating strategy to protect male's paternity (see review by [8]), or as means for female rejection of unwanted, excessive copulations [6], both referred as the 'mating strategy hypothesis' hereafter. Despite some support for the mating strategy hypothesis by behavioral trials in several spiders [12–14, 36, 37], counter-examples nevertheless abound [8, 38]. Several genital plugs inserted in one copulatory opening found in some nephilid spiders indicate that these plugs fail to impede female remating [6]. There are also cases where mating plugs are formed collaboratively from both sexes [7, 33, 39]. Furthermore, secretion plugs in some spiders not only block copulatory openings, but also the groove slits on the epigynal surface, even the whole epigynal plate (e.g., *Neriene emphana* (Linyphiidae) [7]: Fig 1D; *Eresus kollari* (Eresidae) [40]: Fig 4; *Leucauge argyra* (Tetragnathidae) [41]: Fig 48A; *Metleucauge eldorado* (Tetragnathidae) [41]: Fig 79A; *Tengella radiata* (Tengellidae) [42]: Fig 171D; *Toxopsiella minuta* (Cycloctenidae) [40]: Fig 172A). These imply that mating plugs may also take other functions beyond being an obstacle for mating. Considering the lack of general evidence for the paternity protection function [43, 44], Huber [11] proposed an alternative hypothesis, that plugs might function to prevent the sperm lodged in the spermatheca from leakage and desiccation, referred as the 'sperm protection hypothesis' hereafter. So far, the only empirical evidence for Huber's sperm protection hypothesis is attributed to the case of *Neriene emphana*, a linyphiid spider [7]. However, the genital morphology and mating behavior of *Neriene* species are quite unique [45], atypical of entelegyne spiders. Thus, the sperm protection hypothesis has yet to be broadly tested.

Of course, the mating strategy and the sperm protection hypotheses need not be mutually exclusive, particularly considering different plug types. Broadly speaking, the mating strategy should only apply to polyandrous species, but a sperm protection mechanism matters to all entelegyne spiders. To serve as a sperm protection mechanism, mating plugs should be lodged at spermathecal entrances, while as a mating strategy, mating plugs should present a physical obstacle to sperm entering spermathecae anywhere within the copulatory tract. External examination is oftentimes difficult and reveals very little of whether spermathecal entrances are blocked. Behavioral tests are also deficient as they may reveal that plug presence significantly decreases female remating rates, but cannot at the same time test for their role in sperm protection. Given that mating plugs have in fact been documented to reach spermathecal entrances, e.g., genital plugs in [46–48]; secretion plugs in [10, 41], they could plausibly serve both functions at the same time. Whether these plugs have originally evolved for one function, then taken the other, is unknown.

We believe that a careful morphological study of the details of inner genitalia might begin to clarify these unsolved mysteries. The epigynal tracts of a tetragnathid spider *Diphya wulingensis* provide an ideal model system to test whether mating plugs might serve as a remating prevention, or as a sperm protection mechanism. The spermathecae in *D. wulingensis* have two chambers with the entrance and the exit located oppositely [16]. As many spiders, the blind ball-shaped sac adjacent to copulatory opening in *D. wulingensis* is the secondary spermatheca, while the one linking to fertilization groove is the primary spermatheca [42]. Specifically, paired slits connecting between copulatory openings and fertilization grooves extend along the epigynal plate parallel with the tracts connecting between the two spermathecal chambers beneath the epigynal plate [16]. Such characters imply the spermathecal tracts connecting between the two spermathecal chambers might be in a groove state and the slits are their split openings on the epigynal plate [49], but see [42, 50]. If so, one may expect that mating plugs lodged at different sites might have different biological significance: blocking copulatory openings should impede female remating, but to protect sperm from leakage, desiccation and backflow, plugs need not only block copulatory openings but also seal the whole spermathecal groove slits. Furthermore, if mating plugs should serve as a sperm protection

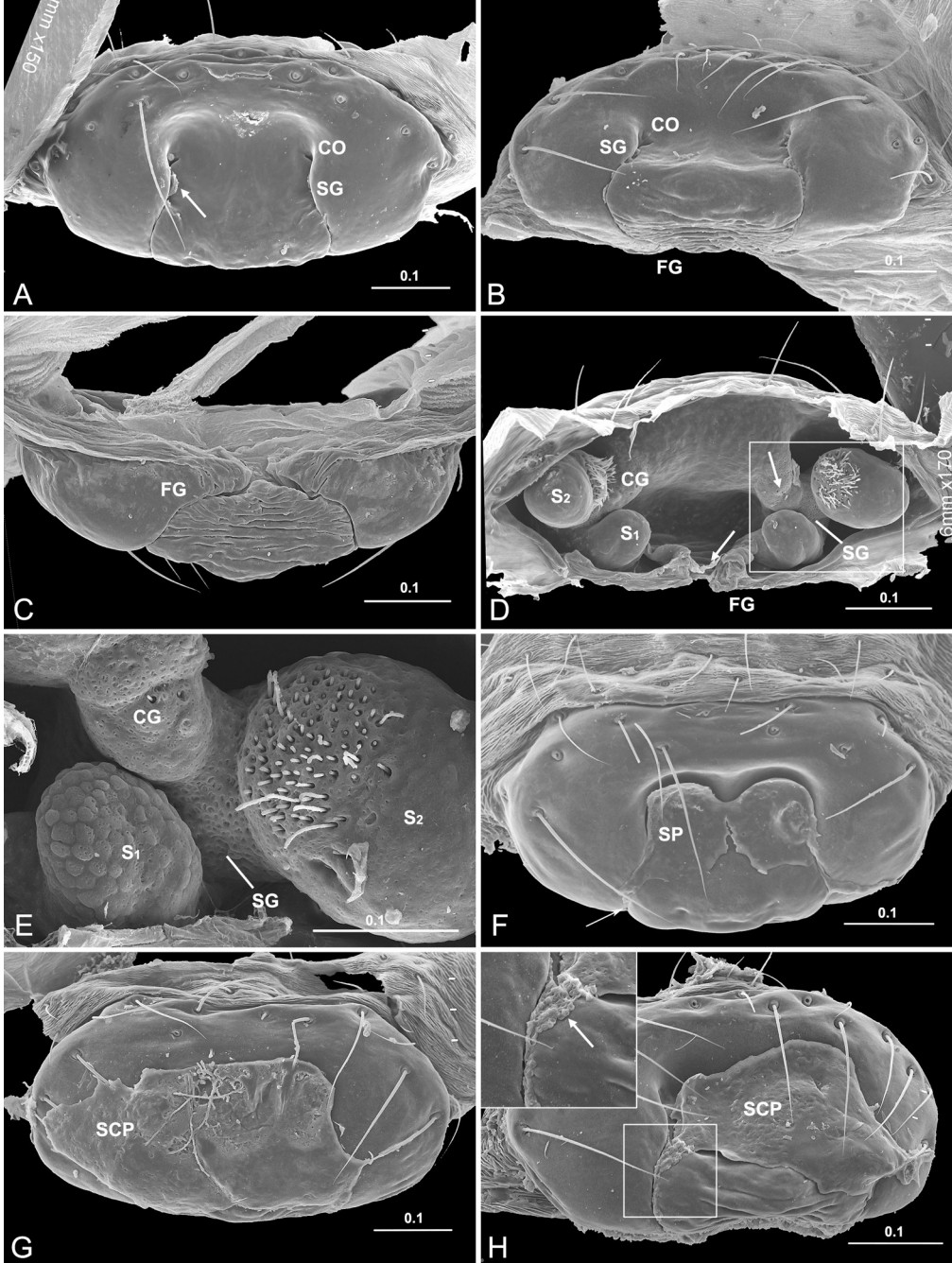

**Fig 1. Epigynum and mating plugs of *Diphya wulingensis*.** A, ventral view, arrow points to fragmented SP in SG; B, caudal view, C, dorsal view; D, inner view, arrows indicate direction of sperm flow; E, detail of D, shows gland pores on $S_1$ and $S_2$; F–H, ventral view, shows mating plugs on epigynal plate, arrow in H points to SP spilled from SG. CG, copulatory groove; CO, copulatory opening; FG, fertilization groove; $S_1$, primary spermatheca; $S_2$, secondary spermatheca; SCP, secretion plug; SG, spermathecal groove; SP, sperm plug. Scale bars: mm.

mechanism, additional problems would persist. For example, any sperm deposited within the secondary spermathecae would need to first transfer into the primary spermathecae, then be released to fertilization grooves. It is difficult to imagine that mating plugs, which would

effectively function for sperm protection, would not at the same time present an obstacle for sperm release and female remating.

We first focused on *D. wulingensis* to test whether mating plugs serve as sperm protection or a mating strategy. We conducted a morphological study on epigynal tracts of *D. wulingensis* by scanning electronic microscope (SEM) and semi thin histological series sections (HSS). We untangled the internal structure of epigynal tract and its relationship with the mating plug by 3D-reconstruction based on a set of HSS pictures. If *D. wulingensis* females are polyandrous, we asked how mating plugs can solve the sperm protection issues without hindering female remating, and how, given that the entrance and exit to its secondary spermatheca share a location on a stalk, the sperm leaves it to fertilize eggs. Due to the immense variation in the morphology of entelegyne spider spermathecae, and because various mating plugs may have multiple origins [8], we then undertook a widely comparative study on spermathecal morphology of entelegyne spiders to test whether there are multiple resolutions to sperm protection.

## Materials and methods

Adult female *D. wulingensis* were collected from Mt. Wuling (N40.56334˚, E117.48740˚, alt. 1112−1176m), Beijing, China on 11−12 Aug. 2009. Epigyna of nineteen females were examined, including seventeen by SEM and two for histological study by semi thin HSS. One of the two sets of HSS pictures was used for 3D-reconstruction. For the comparative study, epigyna were broadly sampled to maximize phylogenetic representations and to emphasize variation in the anatomy of spermathecae. Besides the here examined spiders, information on additional species was collected from published SEM images [41, 51−58]. To check the state of sperm deposited within the spermathecae, four species, *Neriene emphana* (Linyphiidae), *Parasteatoda tepidariorum* (Theridiidae), *Hahnia zhejiangensis* (Hahniidae) and *Neoscona* sp. (Araneidae), were also applied for HSS studies. Furthermore, more individuals of *Neriene emphana* (Linyphiidae), *Acanoides hengshanensis* (Linyphiidae) and *Trichonephila clavata* (Nephilidae) were examined to check the occurrence of mating plugs. The collecting data of the materials examined here is provided in S1 Table. All the materials were preserved in 95% ethanol and stored at −80˚C prior to preparation.

Epigyna were primarily examined using a Leica M205A stereomicroscope. To examine epigynal tracts by SEM, seven epigyna of *D. wulingensis* and that of all other species for comparative study, were prepared as described in [59]. Non-chitinous abdominal tissues were digested with Pancreatin (Sigma LP 1750) enzyme complex, then cleared by an ultrasonic cleaner before drying. The other ten epigyna of *D. wulingensis* used for plug examination were only gently cleared with water to avoid mating plug deformation by enzyme and ultrasonic cleaner. SEM images were taken using a LEO 1430VP in the Department of Biological Sciences at George Washington University and a Hitachi S-3400N at China Agricultural University.

The semi-thin serial sections (1μm) were applied to the spider opisthosoma and performed at China Agricultural University using a Leica EM UC6 microtome with a glass knife and stained with toluidine blue (1%) in an aqueous borax solution (1%) at approximately 90˚C for 1–4 minutes. All slides were examined using a Leica DM5500B light microscope and images were collected with a Leica DFC 500 camera. Information of epigynal structurers was extracted from the HSS pictures and used for 3D-reconstruction. Data were visualized and processed using the 3D analysis software Avizo 9.0. The parameters used to build graphs for *D. wulingensis* include: Input resolution (px): 1700*1700*237; Input voxel size: 1*1*2.37; Filter: Lanczos; Mode: dimensions; Resolution (px): 480*480*237; Voxel size: 3.54167*3.54167*2.37. The main points extracted from HSS pictures include epigynal plate, copulatory grooves, fertilization

grooves, spermathecae, slits of epigynal tracts, and internal spaces of epigynal tracts. All these structures were lined in color on the black-and-white reversal images in Avizo 9.0.

## Results

### Epigynal anatomy

In *D. wulingensis* the epigynum consists of a plate located on the abdomen surface in front of the epigastric furrow, and of a pair of epigynal tracts under it (Fig 1A–1D). The epigynal plate is sclerotized, slightly concaved centrally, and bulged posteriorly. The copulatory openings are located in the centrally concaved area. A pair of slits, starting from the copulatory openings, extends vertically along the epigynal plate and anteriorly into the epigastric furrow. The paired epigynal tracts beneath the integument, consisting of the copulatory tract (CT), the fertilization tract (FT), and the spermatheca (S), accord with a general model of CT-S-FT. The spermathecae in many spiders are globular, but in *D. wulingensis* they are elongate and tubular, and thus resemble tracts, with their entrance and exit located at two ends (Fig 1D). Each spermatheca consists of two spermathecal chambers: a U-shaped tract ($S_1$) and a blind ball-shaped sac ($S_2$), as well as a spermathecal tract (ST) connecting them. The two spermathecal chambers, $S_1$ and $S_2$, traditionally termed as primary and secondary spermathecae [42], possess numerous gland pores on the wall (Fig 1E), and link the fertilization tract with the copulatory tract, respectively (a tract model of CT-$S_2$-ST-$S_1$-FT). Considering the proximity of the paired copulatory openings with the $S_2$ entrances, the CT in this species is extremely short. The paired FT in the inner, dorsal side of the epigynum, connects the $S_1$ and extends convergently into the epigastric furrow.

### Epigynal tracts

Epigynal tracts are integrated with the epigynal plate in *D. wulingensis*. The inner view shows that the tracts are not separated from the epigynal plate (Fig 1D). The cross sections of the epigynal tracts are not closed circles, but have a break opening on epigynal surface, as the integument of epigynal plate continues with the inner surface of the tracts (Fig 2). The tract wall in cross sections is specially modified into a tadpole-shape (Fig 2D and 2E), the bottom cavity opens on the epigynal surface through the "tail" part. Comparatively, the walls of copulatory tracts and spermathecae are as thick and strongly sclerotized as the integument of the epigynal plate, while the walls of fertilization tracts are much thinner, less sclerotized (Fig 2F–2H), and are easily broken during specimen preparation (Fig 1D). In addition, the two chambers and the duct between them in *Hahnia zhejiangensis* (Hahniidae) have sperm inside (S1D and S1E Fig) that indicate both of them are spermathecal chambers, and the duct between them are spermathecal ducts, rather than copulatory ducts as usually recognized (but see [60]). No evidence shows that the sperm deposited in the spermathecae would be encapsulated in discrete packages in order to avoid the mixing of sperm from different males (Fig 2C, see also S1 Fig) as reported in some spiders [61].

### 3D-reconstructed epigynal tracts of *Diphya wulingensis*

The 3D-reconstructed epigynal tracts of *D. wulingensis* are "grooves", rather than closed "ducts" (a tract model of CG-$S_2$-SG-$S_1$-FG, hereafter). In the reconstructed model, the groove bottom cavity and the "tail" part, extend through the entire epigynal tract (Fig 3), the "tail" parts join together to form an edge of the grooves, and the "tail" end openings connect to become a pair of slits on the epigynal plate. From the reconstructed model one can see that the structure of the spermathecal groove (SG) consists of groove cavity (GC) and groove edge

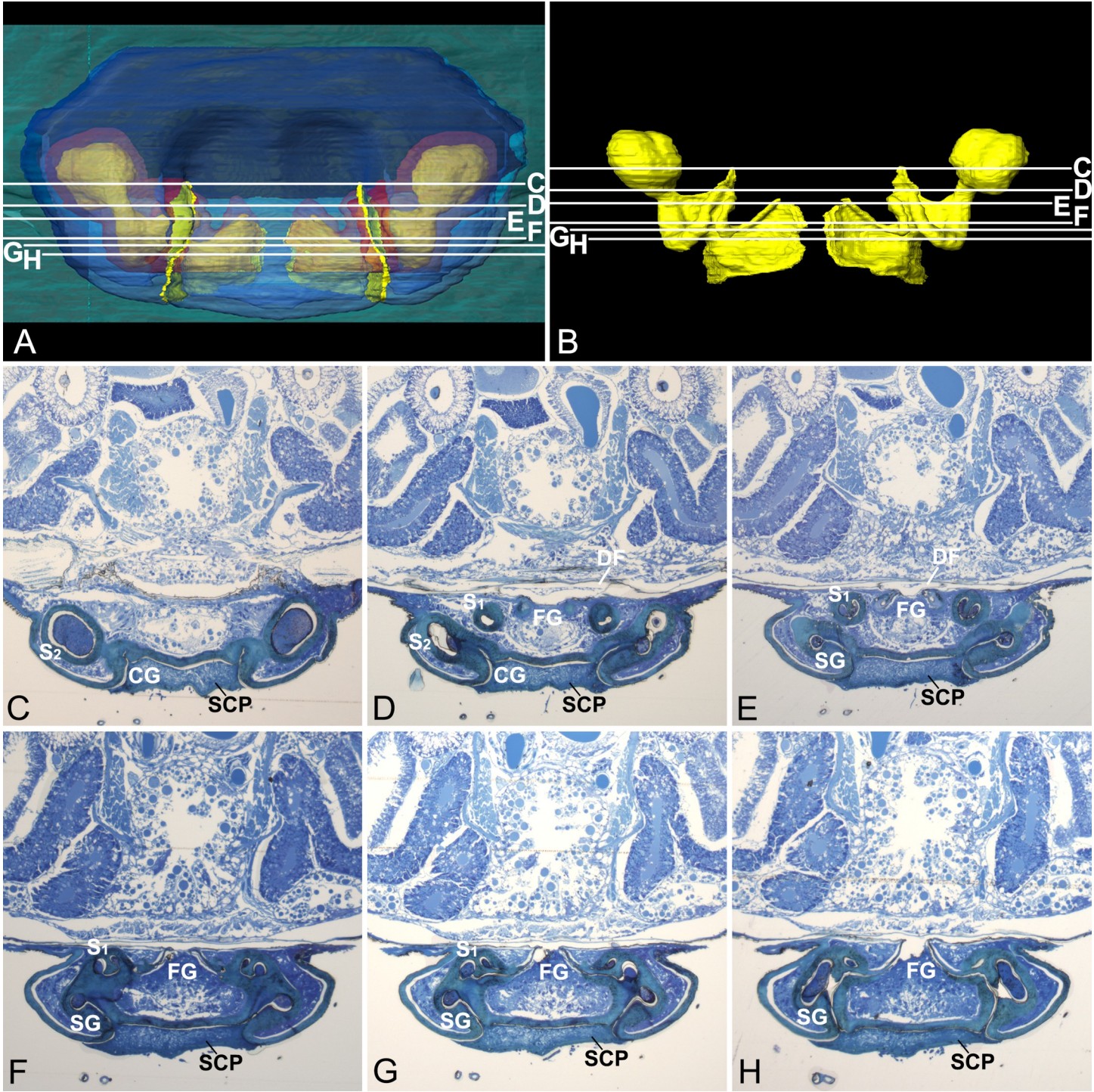

**Fig 2. Cross sections of epigynum of *Diphya wulingensis*.** A, 3D-reconstructed epigynum, ventral view; B, internal space of reconstructed tracts; lines in A−B show section positions of C−H; C, section cross CG; D, section cross $S_2$ stalk, left connecting to CG, right to SG; E, section cross SG; F, section cross SG and $S_1$; G, section cross $S_1$ and FG; H, section cross FG. CG, copulatory groove; DF, dorsal fold; FG, fertilization groove; $S_1$, primary spermatheca; $S_2$, secondary spermatheca; SCP, secretion plug; SG, spermathecal groove.

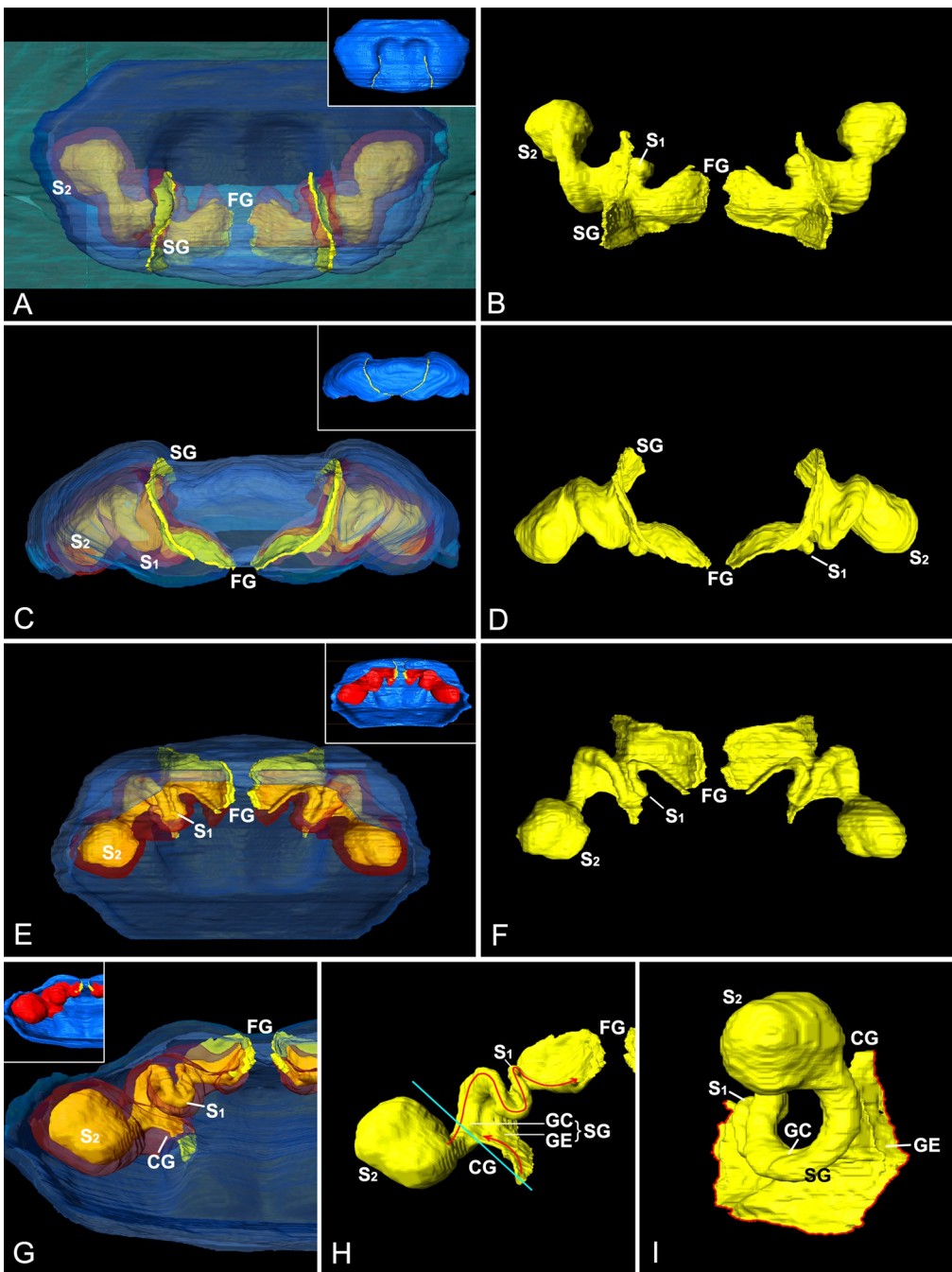

**Fig 3. 3D-reconstructed epigynum of *Diphya wulingensis*.** A, C, E, G, epigynal plate and epigynal tracts, with integument of epigynal plate and tract wall semitransparent, those in thumbnails opaque; B, D, F, H, I, internal space of epigynal tracts; A–B, ventral view; C–D, caudal view; E–F, dorsal view; G–H, inner view, arrows in H indicate direction of sperm flow, blue line shows cross section of $S_2$ stalk; I, lateral view, red line indicates slit opening on epigynal plate. CG, copulatory groove; FG, fertilization groove; GC, groove chamber; GE, groove edge; $S_1$, primary spermatheca; $S_2$, secondary spermatheca; SG, spermathecal groove.

(GE), the two spermathecal chambers: $S_1$ and $S_2$ are derived from the groove cavity (Fig 4). As a U-shaped tract, $S_1$ is formed by a part of groove cavity enlarged into a pocket, with the two integument layers tightly close to each other at the central area (Fig 4A), and laterally folded (Fig 2E), resulting in a U-shaped chamber along the pocket bottom (Figs 1D and 3H and 4A),

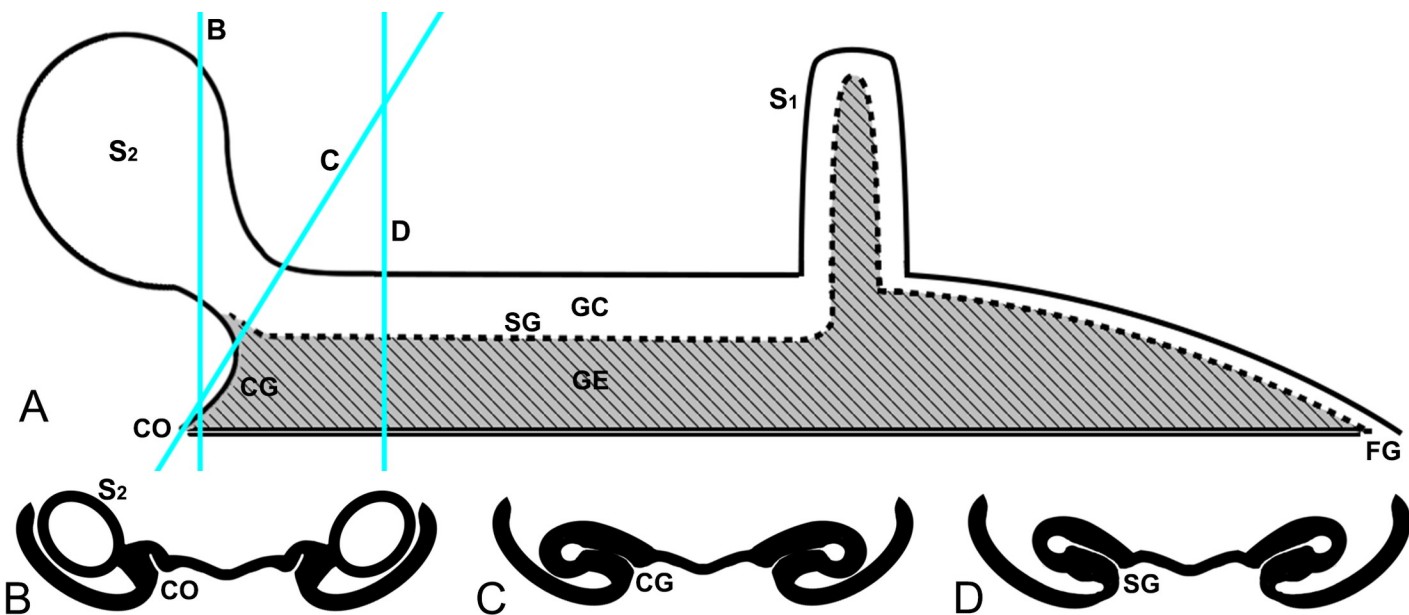

**Fig 4. Schematic structure of epigynal tract of *D. wulingensis*.** A, schematic tract, space represents spermathecal chamber and groove cavity, shadow represents groove edge, double lines indicate groove slit on epigynal plate, blue lines indicate section positions in B–D; B, section cross CG and $S_2$; C, cross section of $S_2$ stalk; D, cross section of SG. CG, copulatory groove; CO, copulatory opening; FG, fertilization groove; GC, groove cavity; GE, groove edge; $S_1$, primary spermatheca; $S_2$, secondary spermatheca; SG, spermathecal groove.

with one end linking to the fertilization groove (Fig 3F). The blind ball-shaped $S_2$ arises from the distal end of the spermathecal groove cavity and resembles a cul-de-sac spermatheca, having the entrance and exit included within the same basic stalk (Fig 1D). However, the inner space of the stalk is also modified into a tadpole-shaped in cross section, having the "tail" part linked to the copulatory groove and the cavity part serving as an exit to the spermathecal groove (Figs 3H and 3I and 4A and 4C). The incoming and outflowing sperm thus do not equally share its internal space. Structurally, the $S_2$ stalk not only docks to the spermathecal groove, but also has its edge in part extend distally to the copulatory opening. Therefore, the copulatory groove is comprised only of the edge part, without a bottom cavity (Figs 3I and 4B), and is extremely short, making the entrances of $S_2$ adjacent to the copulatory openings (Fig 3H). The slits of fertilization grooves open on the epigynal dorsal surface (Fig 3C). Consequently, the paired slits on the epigynal plate (Fig 1A) match the slits of spermathecal grooves connecting between $S_1$ and $S_2$ (Fig 3A and 3I). Besides spermathecal entrances, the groove-shaped spermathecae in *D. wulingensis* also have paired slits, opening on the epigynal plate. Although we did not find any internal structures that might close the spermathecal entrances and groove slits, the modified spermathecal walls prevent the groove cavities to be directly opened on the epigynal plate, but rather via the groove edges which have the two layers tightly closed (Figs 3I and 4I).

## Sperm plug and secretion plug in *Diphya wulingensis*

The amorphous plugs found in *D. wulingensis* vary in size and texture (Fig 1F–1H). The different textures suggest that plug materials are different in their density and liquidity. Two types can be recognized: sperm plug (hereafter SP, Fig 1F) and secretion plug (hereafter SCP, Fig 1G). The SP materials usually contain many rounded sperm-like pellets, nesting within groove slits or flattened on the central concaved area of the epigynal plate, intact or fragmented (Fig 1F and 1H). The hardened scabs are thin and fragile, usually have a flat or slightly concaved

surface that indicates their plug materials are thin liquids with good fluidity. The hardened SCP materials have a bumpy surface and are difficult to be removed by dissection without destroying the epigynal tracts (Fig 1G). These indicate the SCP materials are much denser and harder than those of SP. Some SCP materials also contain rounded sperm-like pellets, but when present, these are covered under thick secretions. This suggests that SCP are added upon the existing SP (Fig 1H). In HSS pictures the SCP materials are different from the sperm deposited within the spermathecae in color and texture, and are separated from the sperm in the groove cavity by the groove edge (Fig 2). In some cases, the entire ventral surface of the epigynal plate can be covered with the SCP, including the central concaved area, as well as the posterior bulged area (complete SCP, Fig 1G). In these cases, the copulatory openings and the paired spermathecal groove slits on the ventral surface are overlaid entirely. In some cases, the SCP is less complete with the plug materials only covering a part of the groove slits, or the slit of one side only (incomplete SCP, Fig 1H). In all these cases, plug materials are only found on the ventral epigynal surface, leaving fertilization groove slits on the dorsal surface free.

Among the 17 epigyna of *D. wulingensis* examined by SEM, five had SP at both sides, fragmented or intact; eight had a complete SCP simultaneously sealing copulatory openings and the slits on the ventral surface, four had a SP on one side and SCP on the other, and in two cases the incomplete SCP did not block the entire groove slits (Table 1 and Fig 1). Both SP and SCP remained intact after being treated by enzymes. However, in one case, after having the epigynum treated by both enzymes and supersonic cleaner, the sperm plug became fragmented with small pieces nested within groove slits, and in another case the large SP scab was broken by tweezers during mounting.

## Four types of entelegyne spermathecae

Besides *D. wulingensis*, 394 epigyna from 330 species, 187 genera, 36 families were also examined here. Based on the locations of the two openings and the architecture of spermathecae, we

**Table 1. Mating plugs in *Diphya wulingensis*.**

| Specimen | Enz | SC | Plug type | Notes |
|---|---|---|---|---|
| No. 01 | | | SP-SP+SCP | incomplete SCP |
| No. 02 | | | SCP | complete SCP |
| No. 03 | | | SP | intact |
| No. 04 | | | SP | intact |
| No. 05 | | | SCP | complete SCP |
| No. 06 | | | SP | broken |
| No. 07 | | | SCP | complete SCP |
| No. 08 | | | SP-SCP | incomplete SCP |
| No. 09 | | | SCP | complete SCP |
| No. 10 | | | SP-SCP | incomplete SCP |
| No. 11 | + | | SP-SCP | incomplete SCP |
| No. 12 | + | | SP | intact |
| No. 13 | + | | SCP | complete SCP |
| No. 14 | + | | SCP | complete SCP |
| No. 15 | + | + | SP | small pieces |
| No. 16 | + | + | SCP | complete SCP |
| No. 17 | + | + | SCP | complete SCP |

Enz, epigyna treated by enzymolysis; SC, epigyna treated by supersonic cleaner; SP, sperm plug; SCP, secretion plug; SP+SCP, SP and SCP coexist; SP-SCP, SP at one side, SCP at the other side.

divided these entelegyne spermathecae into four types (Fig 5 and Table 2): The first broad distinction is between tract-shaped spermathecae with the entrance and exit located at each end, or sac-shaped spermathecae having the entrance and exit closed. The tract-shaped spermathecae usually have two chambers, $S_1$ and $S_2$, at each end, the tract connecting between them also has a slit opening on the epigynal plate, like that in *D. wulingensis* (Fig 1D), we term this type as **spermathecae of elongated groove** (**SEG**, a tract model of CG-$S_2$-SG-$S_1$-FG, Fig 5A and 5B). Such elongated spermathecae may have the slits of spermathecal grooves partly or totally fused, which renders the spermathecal grooves into closed ducts; we term this type as **spermathecae of elongated duct** (**SED**, a tract model of CD-$S_2$-SD-$S_1$-FG, Figs 5C, 5D and 6D). In some cases, the closed ducts are separated from the integument (Fig 5E and 5F). Those sac-shaped spermathecae may arise from the groove bottom cavity with the entrance and exit close to each other, and we term this condition as **spermathecae of compacted groove** (**SCG**, a tract model of CG-S-FG, Fig 5G and 5H, or CD-S-FG, Table 2). Alternatively, those spermathecae usually have one sac-shaped chamber, separated from the groove/duct, and instead have a pair of ducts, CD and FD connected to the epigynal tract; we term this condition as **spermathecae of compacted duct** (**SCD**, a tract model of CD-S-FD, Fig 5I and 5J). Most species examined in this study have the spermathecae of SCG type due to the dense sampling of linyphiids, the spermathecae of the other three types are also common among other spider groups (S2 Table). Variation can be found in some spiders, such as one of the two spermathecal chambers reduced or lost (Fig 6), as well as copulatory and fertilization tracts transform between groove and duct.

## Mating plugs in entelegyne spiders

Mating plugs were examined in 137 of the 411 epigyna, and can be found in all the four types of epigyna (S2 Table). In contrast to genital plug (GP), which was only found in six materials, SP and SCP are more common. In most cases, plug materials were found in the copulatory openings (Figs 6E, 7F and 7L) and the copulatory groove slits (Figs 5A, 5I, 7E and 7I), or in the atria that include both copulatory openings (Fig 7J and 7K). Furthermore, in those taxa with SEG spermathecae and those with SED spermathecae having their spermathecal groove slits partly fused, plug materials also seal the spermathecal groove slits (Figs 5A and 6A), and in some cases even occupy a large area of the epigynal plate (Figs 1G and 6C). In some spiders, more than one type of mating plugs coexist (Fig 7), and in some other cases, the plugs of same type may have different plugging states (Figs 1G, 1H, 7J and 7K). In those having both SP and SCP (Fig 7), comparatively, SP materials look like having been spilled out from the copulatory openings or groove slits, but do not necessary block them (Figs 1F, 5I and 7H), while those of SCP usually seem to be lodged externally (Figs 1G, 1H, 6C, 7C, and7F). However, SCP might have different blocking targets. Unlike that in *D. wulingensis*, SCP in *T. clavata* and *A. hengshanensis* focus mainly on the copulatory openings, as well as in part on the copulatory groove slits (Fig 7C and 7F), but in *N. emphana*, plug materials extend along the copulatory groove slits from the copulatory openings to the spermathecal entrances. These plugs seem to block the spermathecal entrances (Fig 7I).

## Discussion

Entelegyne spiders have to deal with the problems of sperm protection caused by the inherent imperfections of the female spermathecal structure: potential leakage and desiccation of sperm after copulation, and potential backflow when sperm are released. Given the immense diversity of spider forms (well over 50 thousand species, [62]), these problems might have been resolved in different evolutionary ways. We show that *D. wulingensis* possesses groove-shaped

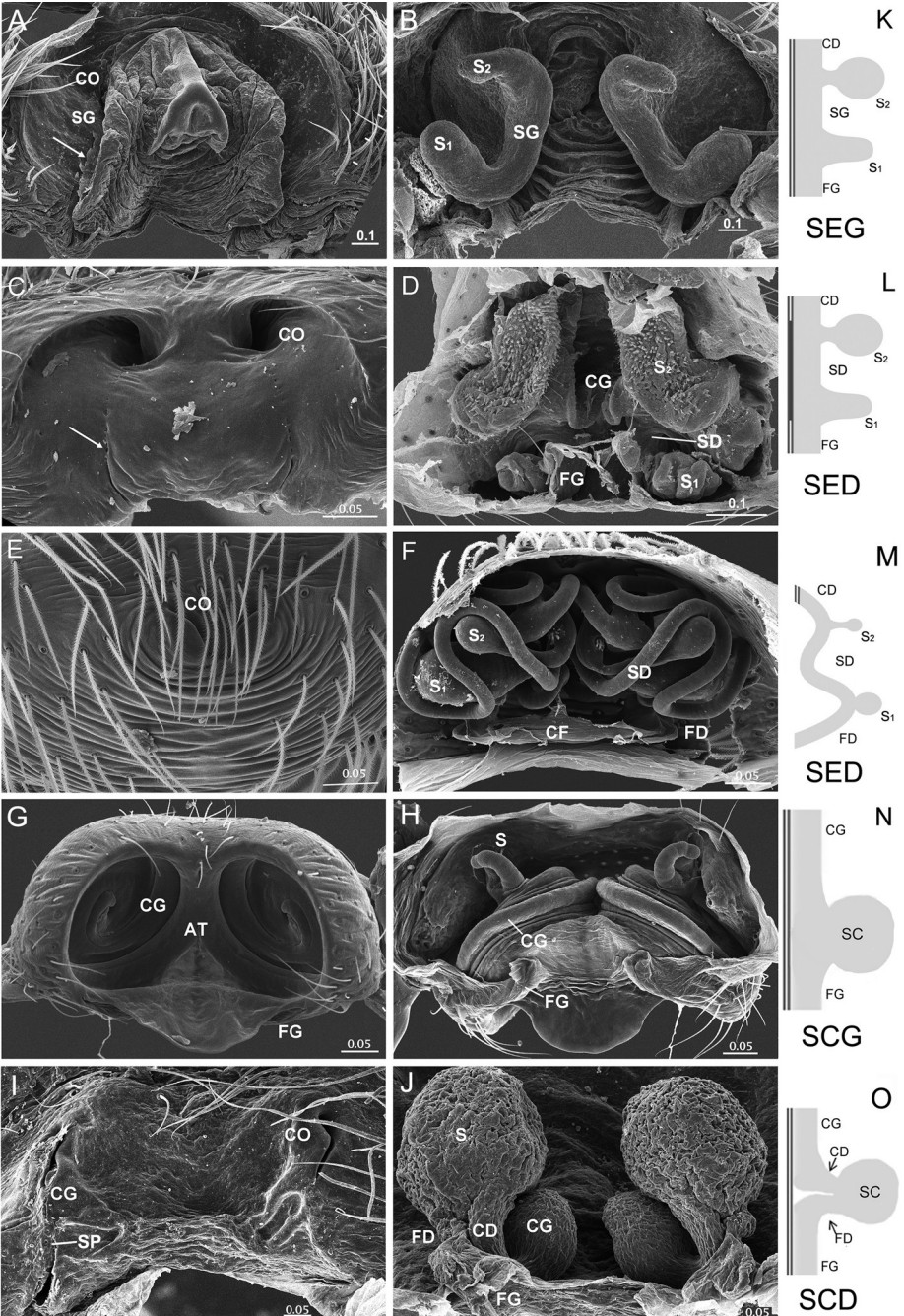

**Fig 5. Types of entelegyne spermathecae.** A, B, K, Type-I SEG, arrow in A points to SP in SG slit; C, D, L, Type-II SED, arrow in C points to fusing trace of SG slit on EP; E, F, M, Type-II SED, without fusing trace on EP; G, H, N, Type-III SCG; I, J, O, Type-IV SCD. A–B, *Tamgrinia alveolifen* (Agelenidae); C–D, *Leucauge* sp. (Tetragnathidae); E–F, *Hahnia zhejiangensis* (Hahniidae); G–H, *Neriene* sp. (Linyphiidae); I–J, *Trichonephila clavata* (Nephilidae); A, C, E, G, I, epigynum ventral view; B, D, F, H, J, same, inner view; K–O, schematic drawings of four spermathecal types. AT, atrium; CD, copulatory duct; CF, common FD; CG, copulatory groove; CO, copulatory opening; FD, fertilization duct; FG, fertilization groove; EP, epigynal plate; S, spermatheca; $S_1$, primary spermatheca; $S_2$, secondary spermatheca; SCD, spermatheca of compacted duct; SCG, spermatheca of compacted groove; SD, spermathecal duct; SED, spermatheca of elongated duct; SEG, spermatheca of elongated groove; SG, spermathecal groove; SP, sperm plug. Scale bars: mm.

**Table 2. Spermathecal types in entelegyne spiders.**

| S-type | ET-style | Species name | Figure | ET characters | Plug on EP |
|---|---|---|---|---|---|
| SEG | CG-SEG-FG | *Diphya wulingensis* | Fig 1 | tract-shaped spermathecae with paired SG slits on EP | EPSCP/ EPSP |
| SEG | CG-SEG-FG | *Tamgrinia alveolifera* | Fig 5A, 5B and 5K | tract-shaped spermathecae with paired SG slits on EP | SGSP |
| SED | CD-SED-FD+CFD | *Hahnia zhejiangensis* | Fig 5E, 5F and 5M | tract-shaped spermathecae having paired CO on EP | — |
| SED | CD-SED-FG | *Leucauge* sp. | Fig 5C, 5D and 5L | tract-shaped spermathecae with fusing traces of SG slits on EP | — |
| SCG | CG-SCG-FG | *Neriene emphana* | Fig 5G, 5H and 5N | paired CG slits on ventral side of EP and paired FG slits on dorsal side | CGSCP |
| SCD | CGD-SCD-FD +CFD | *Parasteatoda tepidariorum* | Zhan et al. 2019: Fig 1E–1G | sac-shaped spermatheca having CD and FD for sperm income and outcome respectively, paired FD convergent into CFD | — |
| SCD | CGD-SCD-FDG +CFG | *Trichonephila clavata* | Fig 5I, 5J and 5O | sac-shaped spermatheca having CD and FD connecting to CG and FG respectively, paired FG convergent into CFD | CGSP/ COSCP |

Epigynal elements: CD, copulatory duct; CG copulatory groove; CGD, CG+CD; CO, copulatory opening; EP, epigynal plate; ET, epigynal tract; FD, fertilization duct; FG, fertilization groove; FDG, FD+FG; CFD, common FD; S, spermatheca; SG, spermathecal groove.

S-type: SCD, spermathecae of compacted duct; SCG, spermathecae of compacted groove; SED, spermathecae of elongated duct; SEG, spermathecae of elongated groove.

Mating plug type: SCP, secretion plug; SP, sperm plug.

Plug on EP, mating plug lodged on epigynal plate; COSCP, SCP in CO; CGSCP, SCP in CG slits; CGSP, SP in CG slit; EPSCP, single SCP on EP coving over both CO and SG slits; EPSP, SP on EP; SGSP, SP in SG slit.

spermathecae (SEG type) having a paired slit-like openings on the epigynal plate and two chambers, $S_1$ and $S_2$ located at each end. The base of $S_2$ has its entrance and exit within the same stalk and close to the copulatory opening, and $S_1$ links to the fertilization groove. Such structural characteristics require a corresponding sperm protection mechanism in the shape of an amorphous plug, not only to block the spermathecal entrances, but also to seal the spermathecal slits. Furthermore, these plugs would not hinder the release of sperm from $S_2$, and would not impede the female's subsequent mating. We found two types of mating plugs in *D. wulingensis*, and their different texture and morphology allow us to label them as sperm, and as secretion plugs, respectively. Our results suggest that a complete secretion plug might serve as a permanent protection mechanism in postcopulation, while the modified spermathecal wall together with the spilled sperm form a temporary protection mechanism to prevent sperm from leakage and desiccation intermating. Furthermore, with the modified spermathecal wall of $S_2$ stalk, the problems of shunt for sperm input and output have been resolved, while at the same time allowing for female polyandrous mating possibilities.

The reconstructed spermathecae of *D. wulingensis* show that a sperm protection mechanism is necessary. As in many entelegyne spiders with the SEG type spermathecae, this spermatheca consists of two chambers, $S_1$ and $S_2$, associating with the fertilization tract and the copulatory tract, respectively (Fig 3). Our results show that the epigynal tracts are not separated from the epigynal plate (Fig 1D), their cross sections are not closed circle (Fig 2), and the breaks in the reconstructed model connect to become a pair of slits on the epigynal plate (Fig 3). These indicate that the epigynal tracts are in a groove state [49]], following a model CG-$S_2$-SG-$S_1$-FG. The paired slits on the epigynal plate (Fig 1A) are not superficial sutures as previously recognized [42, 50], but are rather slit-openings of the epigynal grooves. We have not found any internal structures that might close the spermathecal opening and groove slits (Figs 2 and 3). Similar circumstance can also be found in *Stegodyphus* spiders (Eresidae) [40, 58]. Given that the copulatory grooves are extremely short, and that the fertilization grooves open on the dorsal surface, the paired slits on the ventral surface are largely the openings of the spermathecal grooves. Their walls are thick and strongly sclerotized without any internal closing mechanism to prevent sperm leakage. The fact that SP is found in many other spiders (Figs 5I,

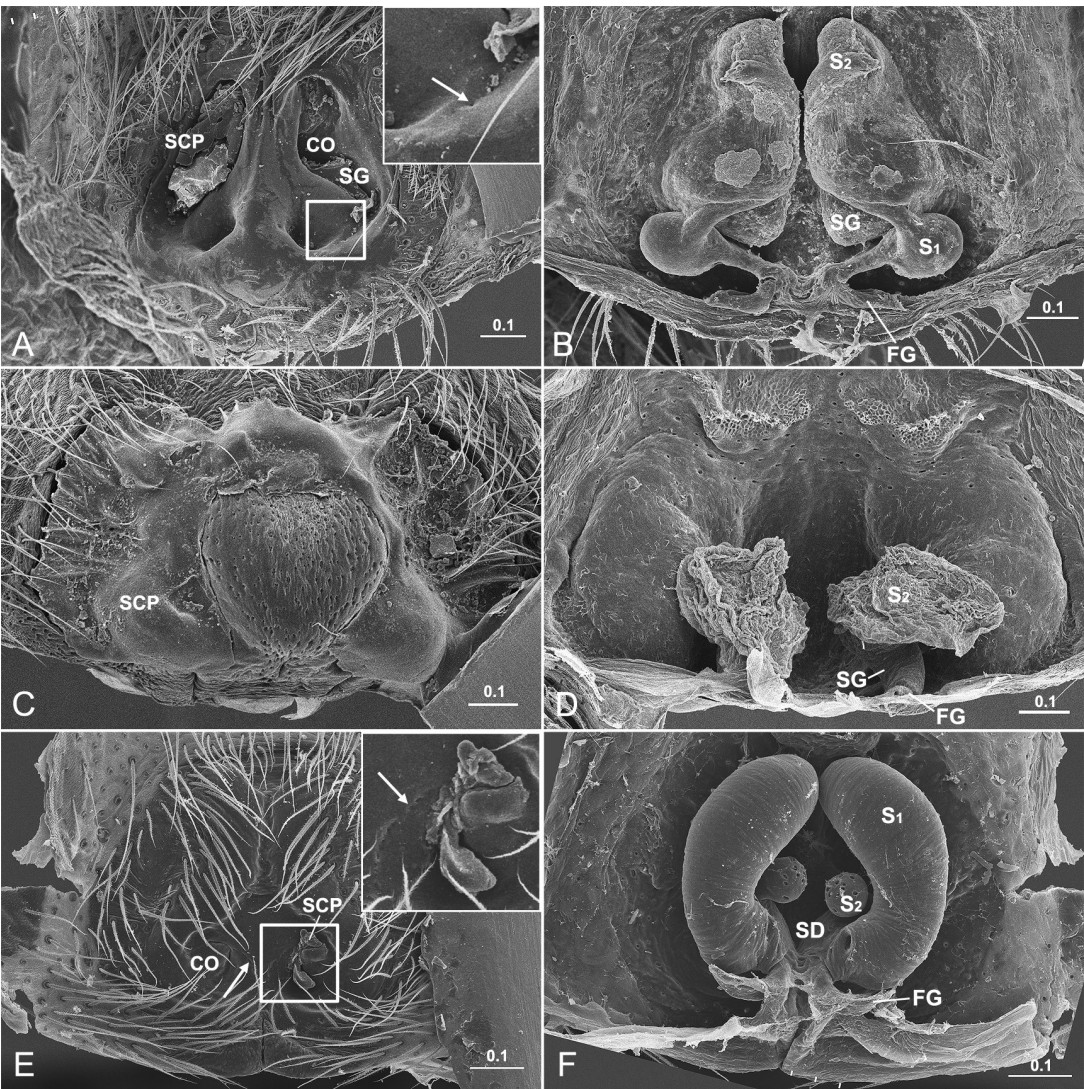

**Fig 6. Variation in elongated spermathecae of entelegyne spiders.** A–B, *Nomisia ausserer* (Gnaphosidae), shows $S_2$ without significant lumen, arrow points to fusing trace of SG slit; C–D, *Metleucauge yunohamensis* (Tetragnathidae), shows $S_1$ lost, $S_2$ as the only significant receptacle; E–F, *Kishidaia albimaculata* (Gnaphosidae), shows well developed $S_1$ and small $S_2$, arrow points to fusing traces of SG slits. CO, copulatory opening; FG, fertilization groove; $S_1$, primary spermatheca; $S_2$, secondary spermatheca; SCP, secretion plug; SD, spermathecal duct; SG, spermathecal groove; SP, sperm plug. Scale bars: mm.

7I and S2 Table) indicates that sperm leakage is a common problem in spiders. In *D. wulingensis* both SCP that covers the entire epigynal plate, and SP nested within the spermathecal slits, are in line with the prediction of the sperm protection hypothesis.

A complete SCP might serve as a long-lasting mechanism for sperm protection. Given the extremely short copulatory grooves in *D. wulingensis*, the SCP plugs lodged in copulatory openings also block the spermathecal entrances (Fig 1D). That a complete SCP covers both the copulatory openings and spermathecal slits (Fig 1G) indicates its function beyond merely impeding female remating. Furthermore, formed by thick secretions, SCP is solid and durable, and is difficult to be experimentally removed. These characteristics suggest that SCP meets the expectations of a sperm protection mechanism that can function permanently. By forming a physical barrier to sperm insemination at the same time, SCP does, however, also lower the

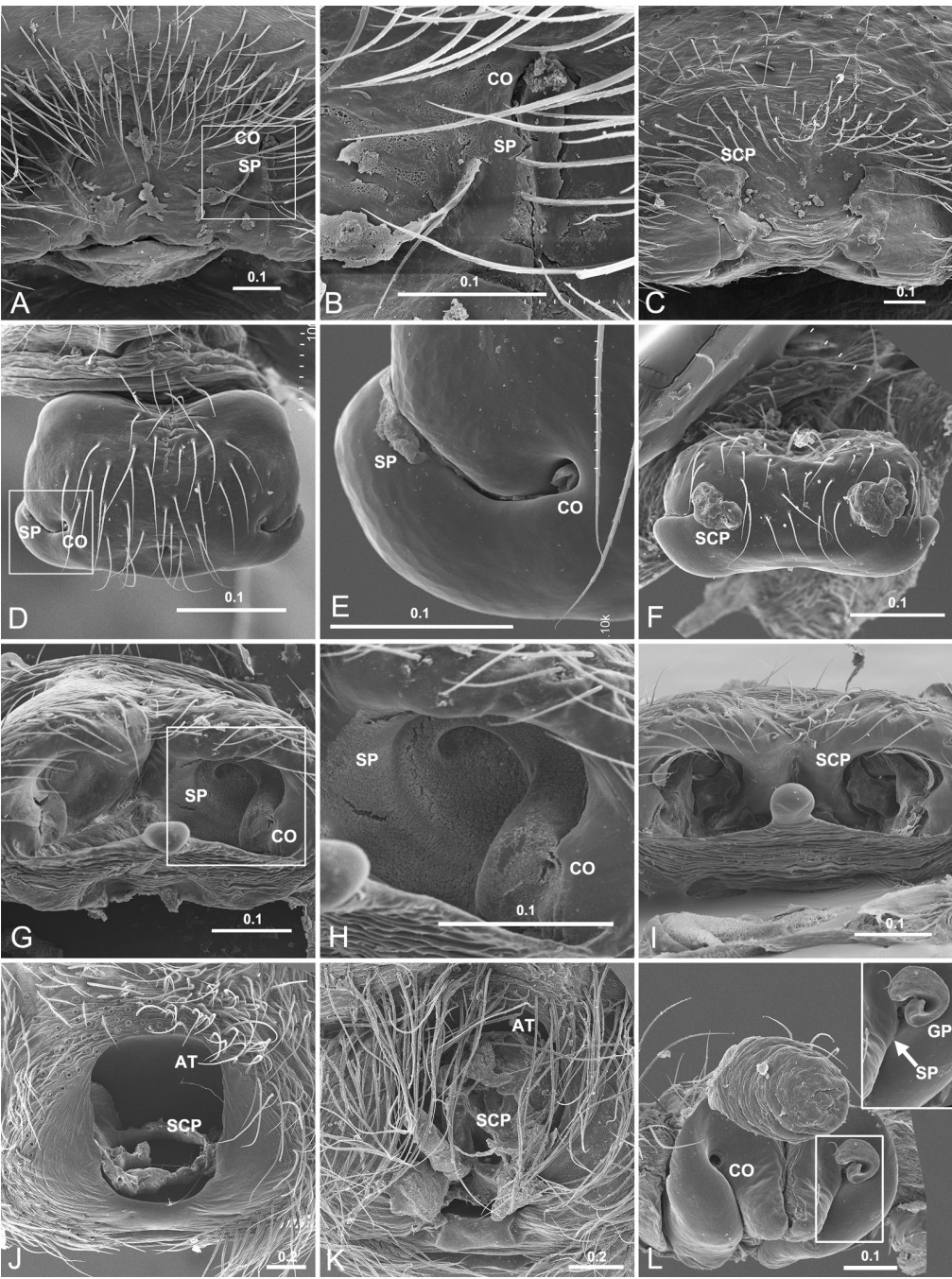

**Fig 7. Mating plugs in entelegyne spiders.** A–C, *Trichonephila clavata* (Nephilidae), showing SP in A–B, SCP in C; D–F, *Acanoides hengshanensis* (Linyphiidae), showing SP in D–E, SCP in F; G–I, *Neriene emphana* (Linyphiidae), showing SP in G–H, SCP in I; J–K, *Agelena sylvatica* (Agelenidae), showing SCP blocking incompletely in J and completely in K; L, *Araneus diadematoides* (Araneidae), showing GP and SP to coexist. AT, atrium; CO, copulatory opening; GP, genital plug; SCP, secretion plug; SP, sperm plug. Scale bars: mm.

possibility of subsequent mating of the female. The texture of the incomplete SCP reveals a thick and well solidified material whose external formation on the epigynal plate (Fig 1H) cannot be accidental, but suggests it be rather intentional. However, our data show that not every mating results in SCP formation (Table 1): the common presences of mating plugs, either SP

or SCP, indicate that all the females examined had mated; eight of them had a complete SCP (8/17), while the others had an incomplete SCP or only a SP. This suggests that the polyandrous mating system in *D. wulingensis* is not jeopardized by plug formation. It thus seems that SCP is responsible for postcopulatory sperm protection.

Our data imply that the modified wall of spermathecal groove together with the spilled sperm form a temporary sperm protection mechanism in *D. wulingensis*. While grooved epigynal tracts have been reported in some entelegyne spiders [42, 49, 63–67], most of them are relevant to the groove-shaped copulatory and fertilization tracts. Nevertheless, *D. wulingensis*, as well as those spiders also having the SEG spermathecae, have also grooved spermathecae (Figs 1A and 5A; see also [58]: Figs 1–3; [68]: Figs 38–41; S2 Table). As stated above, potential sperm spills from the slits, be it post copulation or intermating, would not be surprising given the vagaries of the entelegyne system. Spilled sperm liquid captured between the two edge layers of the specifically modified spermathecal wall (Figs 2E, 3I and 4D) may solidify under surface tension thereby providing a temporary seal of the spermathecal slits. We hypothesize that this mechanism prevents the sperm deposited in the spermathecae from leakage and desiccation. We thus conclude that SP in *D. wulingensis* serves as a temporarily sperm protection mechanism during the intermating stage. Similar SP can also be found in other species with the spermathecae of the SEG type (Fig 5A).

These mating plugs, be it SP or SCP, do not prevent sperm to be released from $S_2$ and an existing SP does not constitute an obstacle to the mating of subsequent males. Generally, SP is formed by the spilled sperm liquid left between the two layers of the groove edge that has little effects to the sperm flow through the groove cavity (Fig 4A). The SCP, on the other hand, is formed by the hardened secretions pasted externally onto the epigynal plate (Fig 1G and 1H), although we have no information about where the secretions come from. HSS analyses reveal that the plug material has a different texture from the sperm stored in $S_1$, $S_2$ and the spermathecal groove (Fig 2). This suggests that separated by the edge part, the thick secretions cannot get into the spermathecal cavities via the spermathecal slits. Nevertheless, $S_2$ is a blind sac arising from the groove cavity, and has its entrance and exit share the same stalk (Fig 1D); hence, sperm inflow and outflow must share the same way. It would thus be challenging to install a mechanism only to prevent sperm leakage, and not also sperm released. The reconstructed epigynal tracts show that the internal space of $S_2$ stalk is modified into a tadpole-shape in cross section (Figs 2D, 2E and 4C), having the wide cavity part dock to the cavity of spermathecal groove, while the narrow edge part associates with the edge of the spermathecal groove, and extends distally to the copulatory opening, forming the copulatory groove with only the edge part, without a cavity (Figs 3H, 3I and 4B). This makes sperm intake to, and sperm outflow from $S_2$ differ fundamentally. During mating the male uses its pointed embolus (see figures in [69]) to penetrate the edge of the copulatory groove and to inject sperm directly into the $S_2$. Although sperm liquid may spill out as the embolus is withdrawn, this mechanism facilitates the spilled sperm and the modified spermathecal wall to form a SP sealing, thereby effectively preventing sperm leakage and desiccation. This has little effect on sperm release from $S_2$ to $S_1$ through the spermathecal groove, and further to the secondary uterus externus [16]. Furthermore, being a thin and fragile layer, such a SP does not prevent the embolic insertion of a subsequent male [70]: it therefore does not prevent female polyandry.

The pressures of sperm protection among the four types of spermathecae are different. The fusing marks left on the epigynal plate in some spiders (Fig 5C; see also [41]: Fig 43A; [42]: Fig 167C) indicate that the spermathecal ducts (SED type) might derive from the slits of spermathecal grooves (SEG type) fused. Ontogenetic evidence shows that the development of epigynal tracts in entelegyne spiders starts with a pair of integuments folds (grooves) including $S_1$ and $S_2$ ($S_1$ = "base of spermatheca," and $S_2$ (= "head of spermatheca," [50]). We infer that the

elongated spermathecae of the SEG type might represent a plesiomorphic condition, the SED type is a derived condition. The "accessory bulb" in some spiders [71, 72] is homologous to the "head of spermatheca" [42, 50] and the ducts between $S_1$ and $S_2$ in *Hahnia zhejiangensis* are in fact the spermathecal ducts without slits fusing traces left (SED type, Fig 5E and 5F) rather than copulatory ducts as previously recognized [60]. Such changes effectively eliminate the risk of sperm spilling out from the slits of spermathecal grooves and hence provide a selective advantage. Comparing to the elongated spermathecae, the compacted spermathecae have the spermathecal tracts shortened towards spermathecal entrances and exits close to each other (SCG type, Fig 5G and 5H); having one of the spermathecal chambers reduced or even lost (Fig 6); having the spermathecal chambers getting inwards and becoming sac-shaped spermathecae (SCD type, Figs 5I, 5J and 6J). All these changes are likely adaptations to resolving the problem of sperm spilling out from the slits of spermathecal grooves. Consequently, except the SEG type, the sperm protection mechanism for the other three types of spermathecae only needs to be targeted at the spermathecal entrances. Due to the coexistence of multiple types of spermathecae in many spider groups (Fig 5A–5F and S2 Table), these morphological shifts are likely to have evolved repeatedly, but this needs further exploration.

Considering mating plugs of varying shapes and origins in the vast morphospace of spiders —a hyper-diverse animal group—, we certainly allow for the possibility that different plugs serve different purposes, and we do find more than one type mating plug coexist in some spiders (Fig 7, see also [6]). These can be adaptations for mating strategies, as well as for sperm protection. Besides those plugs that are formed by parts of broken-off male genital parts [8, 47, 48], SCP may also well relate only to mating strategies (Fig 6C, 6F and 6I), since they only block the copulatory openings and parts of the copulatory groove slits. However, we suggest that amorphous plugs have evolved for sperm protection, and not to prevent female polyandry. If the presence of irremovable mating plugs presents a physical barrier to female remating, then perhaps that is its secondary function. In fact, all entelegyne spiders require a means of sperm protection due to the inherent defects of such spermathecal morphology, and the fact that many spiders do not engage in polyandrous mating may give credibility to this explanation. Therefore, in the species where mating plugs do function as a mating strategy [6, 8], future empirical studies should also test for their sperm protection function, and comparative studies might eventually reveal what function evolved first.

## Conclusions

Through detailed morphological comparations, we demonstrate that all entelegyne spiders have to deal with the problems of sperm protection caused by the inherent imperfections of the female spermathecal structure. In *D. wulingensi*s, having SEG type spermathecae and showing female polyandry, the specially modified spermathecal groove wall together with the spilled sperm form SP to function as a temporary protection mechanism to prevent sperm from leaking and desiccating, while the externally lodged SCP functions in postcopulation, both to prevent additional matings, as well as to serve as a permanent sperm protection mechanism. Furthermore, with the modified spermathecal wall of $S_2$ stalk, the problem of shunt for sperm input and output has been resolved, and the SP would not impede the subsequent matings of the female. More than a single type of mating plugs is also present in other spider groups (Fig 6, see also [6]). Our results suggest that mating plugs might not only serve as mating strategy in spiders, as presumed previously, but also as a sperm protection mechanism; different types of mating plugs might function in different ways; and the problem of sperm protection in the spiders of different type of spermathecae might have multiple resolutions. The present study provides empirical evidence for the hypothesis of mating plug serving as a

sperm protection mechanism. This is attributed to the special morphological features of the SEG type spermathecae in *D. wulingensis*. Whether the original function of mating plugs is to prevent female remating or to protect the sperm is difficult to interpret.

## Supporting information

**S1 Fig. Sections cross spermathecae of entelegyne spiders.** A–B, *Neriene emphana* (Linyphiidae); C, *Parasteatoda tepidariorum* (Theridiidae); D–E, *Hahnia zhejiangensis* (Hahniidae); F, *Neoscona* sp. (Araneidae). CD, copulatory duct; $S_1$, primary spermatheca; $S_2$, secondary spermatheca; SD, spermathecal duct. Scale bars: 0.1 mm.
(TIF)

**S1 Table. Collecting data of materials examined in this study.**
(DOCX)

**S2 Table. Epigynal tracts and mating plugs in entelegyne spiders.**
(XLSX)

## Acknowledgments

We would like to thank Jinzhong Fu for his comments on an earlier version of this paper. We also thank Fernando Álvarez-Padilla, Lara Lopardo, Dimitar Dimitrov, and Gustavo Hormiga for their help on collection for SEM images, and Feng Zhang and Yuri M. Marusik for their help in spiders' identifications. We thank these colleagues who facilitated institutional loans: Norman Platnick and Louis Sorkin (American Museum of Natural History), Shuqiang Li (Institute of Zoology, Chinese Academy of Sciences), Yuri M. Marusik (Institute for Biological Problems of the North, Russian Academy of Sciences), Seppo Koponen (Zoological Museum, University of Turku), Andrei Tanasevitch (Institute of Ecology and Evolution, Russian Academy of Sciences), Hirotsugu Ono (Department of Zoology, National Science Museum, Tokyo) and Akio Tanikawa (School of Agriculture and Life Sciences, University of Tokyo).

## Author Contributions

**Conceptualization:** Matjaž Kuntner, Lihong Tu.

**Data curation:** He Jiang, Yongjia Zhan, Qingqing Wu, Lihong Tu.

**Formal analysis:** He Jiang, Yongjia Zhan, Qingqing Wu, Lihong Tu.

**Funding acquisition:** Huitao Zhang, Matjaž Kuntner, Lihong Tu.

**Investigation:** He Jiang, Yongjia Zhan, Qingqing Wu, Lihong Tu.

**Methodology:** Yongjia Zhan, Huitao Zhang, Lihong Tu.

**Project administration:** Lihong Tu.

**Software:** Huitao Zhang.

**Visualization:** He Jiang, Huitao Zhang.

**Writing – original draft:** Lihong Tu.

**Writing – review & editing:** He Jiang, Matjaž Kuntner, Lihong Tu.

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
