## [Decision Letter · Decision Letter 0]

27 Sep 2023

PONE-D-22-34374A spider mating plug functions to protect spermPLOS ONE

Dear Dr. Tu,

Thank you for submitting your manuscript to PLOS ONE. After careful consideration, we feel that it has merit but does not fully meet PLOS ONE’s publication criteria as it currently stands. Therefore, we invite you to submit a revised version of the manuscript that addresses the points raised during the review process.

We look forward to receiving your revised manuscript.

Kind regards,

Myeongwoo Lee, Ph.D.

Academic Editor

PLOS ONE

Journal Requirements:

Reviewers' comments:

Reviewer's Responses to Questions

**Comments to the Author**

1. Is the manuscript technically sound, and do the data support the conclusions?

Reviewer #1: Partly

Reviewer #2: Yes

2. Has the statistical analysis been performed appropriately and rigorously? 

Reviewer #1: N/A

Reviewer #2: No

3. Have the authors made all data underlying the findings in their manuscript fully available?

Reviewer #1: Yes

Reviewer #2: Yes

4. Is the manuscript presented in an intelligible fashion and written in standard English?

Reviewer #1: Yes

Reviewer #2: Yes

5. Review Comments to the Author

Reviewer #1: This manuscript includes a large amount and detailed work analyzing morphological structures on female genitalia in spiders. On the one hand, authors study in detail the morphology of epigynal tracts in the spider Diphya wulingensi. They found two types of mating plug: sperm plug and secretion plug, and suggested these plugs serve for different functions. The sperm plugs function as temporary protection mechanism to prevent sperm from leaking and desiccating, whereas the secretion plug act for permanent protection mechanism but also for preventing additional matings. On the other hand, they perform a comparative study on spermathecal morphology using several spiders to test whether there are multiple resolutions to sperm protection.

I think the manuscript is very valuable and has an interesting and broad evolutionary approach. However, my first and main concern is that I was not really convinced on the idea of the plug as sperm protection. Authors do not present direct evidence on how sperm is protected by the plugs. I understand morphologies can give us indirect evidence to postulate functions but the actual role of the plugs need to be tested using physiological and behavioral approaches. As far as I followed the text, there is no references for the main problem exposed here in relation to the potential leakage and desiccation of sperm after copulation. I think this information and examples on how sperm is “unprotected” needs to be at the first line of arguments. In relation to the focal species there is lack of information on whether female mate multiple and how the plug affects female willingness to remate, as well as males pedipalp insertions and mating duration. Some kind of theoretical construction around this issue needs to be done in the introduction to better understand the connection with the morphological findings on the study species. Finally, I need to admit that it was absolutely tedious to read the results and follow the main idea tested, so I suggest to revise the results section in order to improve clarity for readers. For the comparative study across different species, I was expecting to see a phylogeny with the different mating plugs, which it would help to see the topic in broader way.

Main comments

Introduction

I think this section needs more work. I suggest to re-write it, giving evidence on sperm viability, mobility and endurance along time in spiders. This can help to build arguments on the problem of dissection. Further, information on sperm leaking is needed.

Also, this section needs to have a paragraph on the sexual behavior of the focal species.

In relation to the main hypothesis, What about if the very large and extensive plugs (here called SCP) found in some species are subject of intense sexual selection?. I can imagine that males can leave a sperm plug (SP) under low sperm competition and additionally a SCP under high sperm competition. What happens with the spider species that don’t have mating plugs? Are these species having less sperm for fertilization? In my opinion, these possibilities need to be discussed at the same level than the sperm protection hypothesis.

Results. This section can be improve with subheadings

Discussion.

This section can be improved following the main suggestion on the introduction.

Minor comments

L30-49. The abstract lacks explicit information on the comparative study

L88-90. These are very odd figure numbers and I couldn’t find the figures. It doesn’t need to be here if the information is from the published papers.

L94-98. Develop the main findings from Huber’s paper.

L152. Here you stated 19 females but in L280 you sated 17 females. Please revise it.

L192. I suggested the subheading (or similar): Female genital anatomy and sperm storage

L194-255. This first section is descriptive, which is fine but very tedious. I suggest incorporate information on the function of the structures to help readers to follow it.

L259. I suggested the subheading (or similar): Sperm and secretion plugs

L280-287. This information needs to be at the beginning of the section.

L289. Table 1. What is the difference between complete and intact?

L291. I suggested the subheading (or similar): Copulatory plugs along the phylogeny

L291. Here you stated 394 epygina from 330 species, but in L317 you stated 137 out of 411 mating plugs. How did you reach these numbers? I think it is important to put all the information together to understand the sample sizes.

L291-312. Please indicate the name of the epygina type in bold

L318. This is the first time you mentioned genital plug in the text. What is the difference between mating and genital plug? Before you defined sperm plug and secretion plugs, I am confused.

L291-334. It is important to include the phylogeny and map the type of mating plugs.

L338-340. Is there any evidence showing that these problems exist?

L341. Please add a reference on the amount and diversity of spiders

L387. I don’t think you can give an intentionality

L388. Did you perform matings? how do you know females were mated?

L409-410. How do you know?

L426-433. This information needs to be placed before

L464-465. I don’t understand this sentence.

Reviewer #2: Hello to all the authors,

The manuscript documentation is of excellent quality, in particular the histology cuts and 3D reconstructions. I consider that the manuscript is publishable. The following ideas are optional and I think will make the paper even better.

1)The authors talk about testing the hypothesis whether SP and SCP could have evolve for sperm protection in addition to prevent sperm competition. The anatomical description of the different genitals systems is very thorough and useful, but it only focuses in spiders that usually present plugs. Do you consider a descriptive comparison with epigyna of species that do not present them is relevant?

2) In the S2Table you include species that present plugs and species that do not in addition to the types of genital systems described. Do you consider useful to apply some statistical analysis with these data? I think that will give more support to the testing part of the paper.

3) Do you consider relevant to mention volumes of the different spermathecal cavities, ducts or groves? This of course depend how difficult is to get these data. Is this data possible to obtain from the 3D reconstructions?

I hope these comments help to improve your manuscript.

Best regards

Fernando Álvarez-Padilla

6. PLOS authors have the option to publish the peer review history of their article (what does this mean?). If published, this will include your full peer review and any attached files.

Reviewer #1: No

Reviewer #2: **Yes: **Fernando Álvarez-Padilla

---

## [Author Response · Author response to Decision Letter 0]

8 Jan 2024

Thanks for the comments of the two reviewers. We have revised the MS following the comments of reviewers. Hope this revised version meets the requirements for publication on PLoS ONE. For details please find in the file of 'Respond to reviewers'.

---

## [Decision Letter · Decision Letter 1]

16 Feb 2024

PONE-D-22-34374R1A spider mating plug functions to protect spermPLOS ONE

Dear Dr. Tu,

Thank you for submitting your manuscript to PLOS ONE. After careful consideration, we feel that it has merit but does not fully meet PLOS ONE’s publication criteria as it currently stands. Therefore, we invite you to submit a revised version of the manuscript that addresses the points raised during the review process.

Comments from the reviewers are attached. Please address reviewer' minor comments point by point. 

We look forward to receiving your revised manuscript.

Kind regards,

Myeongwoo Lee, Ph.D.

Academic Editor

PLOS ONE

Journal Requirements:

Reviewers' comments:

Reviewer's Responses to Questions

**Comments to the Author**

1. If the authors have adequately addressed your comments raised in a previous round of review and you feel that this manuscript is now acceptable for publication, you may indicate that here to bypass the “Comments to the Author” section, enter your conflict of interest statement in the “Confidential to Editor” section, and submit your "Accept" recommendation.

Reviewer #1: All comments have been addressed

Reviewer #3: All comments have been addressed

2. Is the manuscript technically sound, and do the data support the conclusions?

Reviewer #1: Yes

Reviewer #3: Yes

3. Has the statistical analysis been performed appropriately and rigorously? 

Reviewer #1: N/A

Reviewer #3: N/A

4. Have the authors made all data underlying the findings in their manuscript fully available?

Reviewer #1: Yes

Reviewer #3: Yes

5. Is the manuscript presented in an intelligible fashion and written in standard English?

Reviewer #1: Yes

Reviewer #3: Yes

6. Review Comments to the Author

Reviewer #1: (No Response)

Reviewer #3: Comments for authors

In the present manuscript, the authors test the hypothesis that mating plugs, in addition to being adaptations for mating strategies, may serve as a sperm protection mechanism. The manuscript is really well written and of quality. It is well organized and the data set supports, in part, the discussions and conclusions drawn by the authors. In my opinion, the manuscript deserves to be published after the aforementioned corrections and suggestions.

I see three main problems in the manuscript: The first is that although the writing is really well done and the topic is interesting, I think the authors should put a little more information about mating plugs in other taxonomic groups. Because it is a widely distributed journal, readers in areas other than arachnology may not see interest in the manuscript. I suggest then that the first paragraph of the introduction not go directly to spiders and rather be a very brief summary of mating plug studies in other taxonomic groups.

The second problem has to do with the excess of details in the results and the discussion. I understand the need to detail all the structures of the epigynium because the manuscript is through a morphological approach, but the idea is really lost a bit after two full pages of reading the very detailed description of the structures. I suggest that both the results and the discussion be shortened for the reader's convenience.

Finally, although the morphological approach is really good and the detail and comparison between species is really interesting, we know that they are still indirect evidence of the functions of the structures. I really don't know if the authors performed subsequent experiments with the individuals under laboratory conditions to corroborate the conclusions drawn in the manuscript.

Abstract

I understand that a large part of the study was focused on the species D. wulingensis, but I still feel that part of the main results is missing from the comparison that was made with the other species.

Minor fixes

• Page 3, line32: I think the authors should provide a little more information about mating plugs both in invertebrates (which is where it is best known) and in vertebrates. Because it is a widely distributed journal, readers from areas other than arachnology may not see interest in the manuscript. I suggest then that the first paragraph of the introduction not go directly to spiders and rather be a very brief summary of mating plug studies in other taxonomic groups.

• Page 3, lines 60 and 62, could you please put some references?

• Page 5, line 103: missing parentheses to close the sentence.

MATERIALS AND METHODS

Study species

I feel that some biological data is needed on the study species, D. wulingensis, I suggest placing in materials and methods a "study species" section on what is known about the species.

7. PLOS authors have the option to publish the peer review history of their article (what does this mean?). If published, this will include your full peer review and any attached files.

Reviewer #1: No

Reviewer #3: No

---

## [Author Response · Author response to Decision Letter 1]

4 Mar 2024

In this revised version PONE-D-22-34374_R2, all changes have been made following the comments of the Reviewer 3. We sincerely hope that this revised draft answers the questions and meets the requirements of the reviewer.

For details, please see Response to reviewer comments

---

## [Decision Letter · Decision Letter 2]

14 Mar 2024

A spider mating plug functions to protect sperm

PONE-D-22-34374R2

Dear Dr. Tu,

We’re pleased to inform you that your manuscript has been judged scientifically suitable for publication and will be formally accepted for publication once it meets all outstanding technical requirements.

Kind regards,

Myeongwoo Lee, Ph.D.

Academic Editor

PLOS ONE

Additional Editor Comments (optional):

Reviewers' comments:

Reviewer's Responses to Questions

**Comments to the Author**

1. If the authors have adequately addressed your comments raised in a previous round of review and you feel that this manuscript is now acceptable for publication, you may indicate that here to bypass the “Comments to the Author” section, enter your conflict of interest statement in the “Confidential to Editor” section, and submit your "Accept" recommendation.

Reviewer #3: All comments have been addressed

2. Is the manuscript technically sound, and do the data support the conclusions?

Reviewer #3: Yes

3. Has the statistical analysis been performed appropriately and rigorously? 

Reviewer #3: N/A

4. Have the authors made all data underlying the findings in their manuscript fully available?

Reviewer #3: Yes

5. Is the manuscript presented in an intelligible fashion and written in standard English?

Reviewer #3: Yes

6. Review Comments to the Author

Reviewer #3: I reviewed the improved version of the manuscript and saw that all my comments and suggestions were taken into account. The manuscript was really well written and its results and conclusions are in accordance with the objectives and methodology. The current version of the manuscript in my opinion can be published.

7. PLOS authors have the option to publish the peer review history of their article (what does this mean?). If published, this will include your full peer review and any attached files.

Reviewer #3: **Yes: **German Antonio Villanueva Bonilla

---

## [Editor Report · Acceptance letter]

20 Mar 2024

PONE-D-22-34374R2 

PLOS ONE

Dear Dr. Tu, 

I'm pleased to inform you that your manuscript has been deemed suitable for publication in PLOS ONE. Congratulations! Your manuscript is now being handed over to our production team.

Kind regards, 

on behalf of

Dr. Myeongwoo Lee 

Academic Editor

PLOS ONE